# Interaction Diagrams: Development of a Method for Observing Group Interactions

**DOI:** 10.3390/bs9010005

**Published:** 2018-12-30

**Authors:** Kristina Nestsiarovich, Dirk Pons

**Affiliations:** Department of Mechanical Engineering, University of Canterbury, 20 Kirkwood Ave, Christchurch 8041, New Zealand; kristina.nestsiarovich@pg.canterbury.ac.nz

**Keywords:** team communication, engineering communication, observation, group interactions, boundary object, artifact, team roles, mixed methods, written notes, sociograms

## Abstract

Recording of team meeting’s processes with electronic devices can be problematic because of the invasiveness of the process: issues with privacy; interpretation difficulty with noise or quiet speech; and distortion of participants’ behaviour. There is a need for less intrusive methods. We developed the interaction diagram method by extending the directed graph nature of sociograms to capture the time sequence of events, including the identification of the person, communication behaviour, and duration of interactions. The method was tested on engineering team meetings. Data processing by quantitative and qualitative analysis is shown to be feasible. Several team roles were observed in the engineering context: Initiator; Passive collector; Explorer; Information provider; Facilitator; Arbitrator; Representative; Gatekeeper; Connector; and Outsider. The work provides a graphical representation of the record of the interaction flow during meetings. It does this without needing video recording. It is also an efficient method, as it does not require subsequent transcription or coding. It provides a procedure to quickly analyse communication situations, identify group roles, and compare group activity at different meetings.

## 1. Introduction

It is hard to imagine work in organisations today without regular team meetings. Team communication is an integral part of everyday routine activities because of the growing complexity of the decisions undertaken. Teams comprise multiple members with different characteristics and temperaments, and consequently teams develop a communication style and habits of interaction. To observe these behaviours, it is necessary to record features of the communication.

However, recording of team processes can be difficult. While video and audio recording can provide a rich record of the interactions for subsequent analysis, there are multiple detriments: issues with privacy; difficulty of interpretation because of noise or quiet speech [1]; and distortion of the behaviour of team members [2]. Cameras do not create barriers to productivity, but participants do feel it affects their communication style [3]. Furthermore, more substantial ethics approval processes are required, and this prolongs the preparation stage of a research experiment.

Hence, there is value in developing methods that allow researchers to document key features of team communication with less intrusion than audio-visual recordings. It is also advantageous if a method makes for easier data collection, as existing methods can be laborious both in the preparation and in the post-processes stage. It is especially important for those cases where team members are opposed to giving consent for intrusive forms of recording, e.g., in commercially sensitive cases.

This paper develops a novel method for recording meetings in organisations. We refer to this as an interaction diagram (ID).

### 1.1. What Needs to Be Recorded?

The most important things to record include data and time of the meeting, purpose, the sequence of participants’ turn-taking, and decisions made as a result of meeting discussion. In addition, other aspects of communication may be of importance: non-verbal interactions, emotions, artefacts (boundary objects), and team roles. Exactly what people said may also be of interest, especially for qualitative research purposes. 

An important issue raised in the literature is the process of meeting recall. Which aspects of the meeting do people generally remember well? One of the more comprehensive papers in the field [4] shows that people can remember their own speech and activities very well, together with major topics of the meeting, official roles and seat positions of others, whereas memories about other participants’ performance and details such as dialogues, gestures, time sequence and emotional expression are vague. This appears to be the only recent study on the topic of meeting recall. Evidently, there is a need to pay special attention to a variety of features of meetings.

### 1.2. Tools for Recording

There are different methods of recording communication in meetings: taking written notes or minutes on paper and electronic device; stenograph [5]; recording visible key points; audio and video recording with later possible transcription [6]; phone conferences with telephone recording; automated audio transcription [7]; and speech-to-text software [8]. These approaches often overlap.

Electronic technology can record audio and visual interactions of team meetings. The audio may subsequently be transcribed, and the video characterised by some scheme. The problem is that transcription needs time, and also retrieval of the necessary part from the long audio stream can be problematic [9].

Therefore, many software systems have been developed to integrate audio, video recording, and note-taking. One of the first such systems is a combination of audio recording and tablet for handwriting, called “Filochat” [10]. Another is the “Audio Notebook” that allows users to capture an audio recording and link it to the notes written on paper [11]. A recent example is “Livescribe”—a combination of notebook and smartpen [12] that converts handwritten notes into digital text. The camera on the tip of the pen records gestures generated by a user on a writing surface, and then special software allows the user to find and select a necessary word or phrase in the content (written or audio) [13]. The recorded audio optionally can be synced with the written notes. Such systems may help avoid unnecessary transcribing.

Examples of applications include annotated video to examine interactions among forestry workers [14], sport events [15], educational settings [16], and medical interactions in group therapies [17].

However, the focus of all these methods is on the input device and hardware (paper, digital notebooks, and pens) rather than on real data integration [9]. In addition, these systems can be expensive, and it can be difficult to share information between different applications. Another restriction is that they generally only link paper notes to digital information (audio and video), and not the reverse [9].

Recently, there has been an increase in popularity of speech systems where the software presents speech as synthesised and standardised voice [18]. Such tools can enhance the clarity of speech, and reduce self-consciousness of the speaker, but make voice less personal and less natural. Most of these interfaces can produce audio transcription [19,20]. The “TypeTalker” system can produce transcription as well as edited speech and gesture comments [18].

Shortcomings of these programmes are the time taken to learn them, and possibly the cost. In addition, some personal aspects of speech are lost, such as intonation, emotional expressions, and loudness [18], which may influence the interpretation of dialogues at the meeting. In addition, using speech commenting and speech recognition can be difficult in cases of parallel discussions (when many people speak at once).

Any kind of observation influences the behaviour of team members and potentially distorts the results of the experiment (Monahan and Fisher, 2010 [2]), although recording with electronic devices seems to be more invasive. There are many advantages of manual diagrams on paper: they are less sensitive to technical failures, paper diagrams cannot be lost so easily as electronic charts, and researchers do not need to set up any computer programmes. People can visualise the communication situation quickly and easily [21,22]. In addition, people feel less intimidated by paper than computer recording [21].

### 1.3. Sociogram

There is a place for rapid manual methods that are based on taking written notes. One of the most famous is the sociogram—“a systematic method for graphically representing individuals as points/nodes and the relationships between them as lines/arcs” [23]. It was developed by Moreno and Jennings to analyse group preferences. With the evolution of social network analysis towards software tools and graph theory, the word “sociogram” has been replaced by “graph diagrams” [22].

Many studies were directed at communication mapping during the 1940s–1960s. The concept of communication patterns, which is similar to sociograms, uses lines and symbols. Leavitt [24] in 1950 identified and diagrammed typical patterns of communication (wheel, Y, chain, circle and network). These patterns and the position of the participants in it were correlated with behavioural differences of people [24].

In most of these works, the communication pattern is defined as connectivity between nodes (e.g., [24,25]), thus the diagrams are evaluated quantitatively (number of connections, symmetry, etc.). The communication is presented as an exchange of messages between participants through channels, i.e., a sender–receiver model.

In contrast, the method shown in the current paper differs by using diagrams as a platform for qualitative data, although quantitative data can be extracted too. We were not very interested in the relationship structure of the group at meeting or the position of the team members in the communication network, but rather how people react to others’ behaviour, what words and non-verbal signs are used, what proceeds the communication events at the meetings, and how different situations (e.g., parallel discussion, use of artefacts, and appearance of a new person) change the communication environment in the group. In this sense, we were less interested in the communication pattern, and more in the communication behaviours. The ID method provides a mechanism to represent these behaviours by using diagrams with a time-line.

Nowadays, sociograms and graph diagrams are used in the evaluation of the relationship between people within the context of a particular situation, such as project discussions. They help researchers to visualise communication processes and social links in a particular team. Sociograms allow the combining of data from different sources and use of qualitative and quantitative methods of research [22]. Sociograms continue to be adapted. For example, they have also been used to represent the social relationships between people [26]. Typical applications of sociograms have been to ethnic relationships [27], physical education [28], personal relationships [21], skilled migration (Ryan, et al. 2014 [22]), online communities [29], and medicine [30,31].

There are different types of sociogram based on many criteria: similarities (same location, members of the same group, gender, and age group), social relations (roles and friendship), interactions, and information flow (lines of communication) [32]. Each shape (node) on a sociogram indicates a person or organisational unit. Each line indicates a connection between units.

There have been previous attempts to further develop the design of sociograms. For example, it is suggested that diagrams could be enriched with questions, additional notes for qualitative analysis, or statistical graphs for qualitative analysis [33].

The main limitation is that the geometric constructions used in sociograms may misrepresent elements of the communication. For example, the central place in the graph structure is not necessarily the geographical point in the centre of the group [34]. Another limitation is that sociograms struggle to record communication at the meeting as the method captures the structure of a group and links (relationship) between team members, rather than time and event sequences. Introduction of the time dimension is a key development in the present work.

### 1.4. Summary

Methods involving audio and video recording may ask additional efforts from the researcher to receive consent from people involved in the recording. That makes it harder to receive access to meetings. On the other hand, paper methods such as stenograph, or simply transcribing, are less invasive but require particular expertise (stenograph knowledge, good level of language and speed writing). There is a need to develop methods that can combine the advantages of both methods. Sociograms can be used for recording of information at meetings, but need to be improved by including the timescale and information about easily forgotten items such as participants’ behaviour (tone, the speed of speech, gestures, emotional expressions, etc.).

The purpose of this exploratory study was to develop a methodology for team observations. The objectives for the system were that it should represent multiple interactions between participants in a time-pressured situation (and in time sequence), distinguish between different types of interactions, identify non-verbal behaviour, and provide a mechanism to quantify the number and type of communication events that a person makes.

## 2. Materials and Methods

Our approach was to extend the directed graph nature of sociograms, i.e. the arrows, and use these to show the direction of member-to-member interactions. We were particularly interested in capturing the time sequence of events: who starts the interaction and who picks it up? As part of that, we also wanted to know how long each interaction lasted, and the type of interaction. We created a categorisation of the type of interaction and devised graphical symbols for these. We found that this categorisation could readily be extended to include broadcast transmissions, parallel discussions, use of artefacts (boundary objects), non-verbal behaviour (such as gestures), and repeating communication patterns. We refer to this as an *interaction diagram (ID)*.

To refine the interaction diagram method, we followed five teams of engineering students who were doing a final year project. Typically, there were also one supervisor and one client in every student team. Meetings took place once a week or once per two weeks, for 5–7 meetings.

This investigation included only observations. No audio or video was recorded. The researcher stayed aside taking notes and observing the meeting. 

Ethics approval was obtained from the University of Canterbury Human Ethics Committee (HEC 2017/03/LR-PS), and consent was obtained from all participants.

We used these periods of observation to test and further refine the ID method. We were particularly interested in maximising the capture of non-verbal behaviour, within the constraints of a hand-written processes. We developed several refinements to the basic concept, and these are described in the results below.

Communication events at the meeting were divided into several groups. A *point system* was devised to show the contribution of each participant to the team communication. The number of interactions during the meeting was calculated and used to find the density of communication events for each participant and for the whole team, and to identify team roles.

Qualitative data were collected during the observations. After every observation, interaction diagrams were used to extract data and a journal summary was written. This was followed by a process of markup (underlining typical actions) and defining patterns of behaviour for each participant at the meeting.

## 3. Results

The interaction diagram provides a graphical method of representation of communication between team members. The system is explained below.

### 3.1. Basic Principles of the Interaction Diagram

Interactions diagrams were built on the basis of the sociogram, with the addition of sequence or order of communication interactions. *Interaction* was considered to be every communication turn-taking of a team member at the meeting.

Communication flow at the meeting was divided into several time intervals—*slides*. The slide was one piece of paper, and it changed when it was full of information, or it became hard to read the scheme. In our case, we found that one slide could cover approximately 2–8 min of a meeting, depending on the nature of team interactions.

#### Legend

We developed a legend to categorise the type of interaction. This legend includes graphical symbols:In the right corner, the researcher indicates *the starting time* for every slide and the current topic of discussion*Numbers* represent the sequence of communication interactions (every interaction starts with turn-taking)*Letters* represent the participants of the meeting*A circle* represents a broadcast speech that refers to everybody*Arrows* show the direction of communication*A question sign* represents a question asked by a particular person*Small arrows* near the question mark represent answers and repeating questions (see Figure 1)*Notes* may be written near participant’s letter, about his/her communication style or role*Parallel discussions* are shown as big circles around a particular group of participants (see Figure 3)*Green colour* shows participants, starting time and special marks; *blue colour* shows communication processes and notes about team roles; and *red colour* shows a name or abbreviations of participants*Long monologue speech* is shown as a thick line (arrow or circle)*Repeating patterns* are shown as small lines, as a separate group, with numbers of repeating interactions (see Figure 3)*Solid line* means verbal communication interaction, while dotted line is non-verbal (Figure 2)

Examples of the method used in the investigation can be found in Figure 1, Figure 2 and Figure 3.

Our study followed the communicative approach to the team roles developed by Lehmann-Willenbrock, Beck and Kauffeld [35]. The main idea of this approach is that team roles appear and develop in communication situations and through communication patterns, rather than through the official position of participants in the group [35]. Therefore, roles can be identified in observations and can be changed during the meeting or between meetings.

### 3.2. Interaction Diagrams

#### 3.2.1. Case 1: Simple Communication Situation 

In this situation (Figure 1), two supervisors and four students were discussing problems with model testing at the early stage of a project. This was the team’s second meeting, and in the middle of the discussion time. 

Persons B and C were supervisors, while the others were students. The numbers that follow indicate the sequence of interactions as found on the diagram. 

The slide started at 12:22 when Student D proposed a discussion of model testing by appealing to the whole group (1), and then asked a question to Supervisor B (2). Supervisor B answered this question (3). Supervisor C started their participation by expressing their thoughts (4), and Supervisor B commented on this by appealing to the whole group (5). Then, intensive discussions started between Participants B and D: Supervisor B asked a question (6) and received an answer from the student (7), and then asked something else (8) and received a second answer (9). Student E commented (10), Student D continued the idea (11) (using a laptop, legend: “PC”), Supervisor C asked D (12) and received an answer (13). Special observation (see notes on the diagram): Supervisor B took an active part in the meeting, made plans and organised discussion that defined his team role as “Facilitator”. Student D was characterised in this slide as an active “Team Leader” in this group. Students A and F were passive and did not participate in the discussion at all.

#### 3.2.2. Case 2: Use of Artefacts

The second example shows a variety of communication events. This was towards the end of the third meeting of the group with four participants: three students and one supervisor. It is not the same group as before. The team members were discussing some elements of the project model (scale) and used a whiteboard for graphical representation. This slide started at 12:58 pm.

In Figure 2, note the long rows of speech interactions: one arrow and many numbers. That was done because of saving space on the slide.

The top of this slide shows that a material object (artefact) was used by team members for communication. Such an artefact can be presented as a separate graphic object because during communication it possesses some features of an interlocutor (participants generally look at the artefact while talking and not to the other members). The first arrow from the participant to the artefact shows the person who starts using this object (draws a diagram or shows a physical model). Other participants may say or ask something—represented by the solid line. Alternatively, they may show non-verbal behaviour, represented by a dashed line (generally a sign of agreement, disagreement or misunderstanding that can also be annotated with a question mark).

In the current case (Figure 2), Participant A (Supervisor 1) drew a diagram on the board (Interaction 2), Participant B (student Team Leader) took an active role in the discussion of this chart, whereas Students D and C mostly watched the board showing signs of agreement and understanding (dashed lines and circles as short commenting without being addressed to a particular person).

#### 3.2.3. Case 3: Parallel Discussions

This case illustrates an extreme situation. This involved a meeting with many team members and the complexity of multiple parallel discussions happening over a short time frame. Six team members (fifth meeting, middle of discussion time) were discussing the physical model, and the discussion was intensive.

Figure 3 shows there were six people in the room (different people to the previous cases). The meeting led to a parallel discussion in the group. The group was divided into two subgroups: Participants A (Supervisor 1), B and C in one group and Participants D, E and F (Supervisor 2) in another subgroup. Each subgroup had about ten communication interactions. As both communication discussions were very intensive, it was hard for the researcher to catch all of them. Therefore, small arrows and numbers were used to show the quantity of turn-taking among participants of each group.

First, Student B prepared a model (wavy line)—a physical object that was intensively discussed. Then, Student E added new ideas that were discussed in the whole group again. Later, there was a pause. Then, Member B showed a small artefact (chart) after which discussions divided into two different parts. Participants joined the group in the nearest physical proximity.

Participant D seemed to be interested in both conversations (dotted line means non-verbal interaction, interest) and hesitated which subgroup to choose. However, he finally chose the nearest one (specific reactions of participants on the situation are written above or below the arrow).

The challenge in this type of situation is to follow both conversations and do not lose the information. Parallel discussions make observations difficult. Apparently, eight people at the meeting may be a practical maximum for this method. Over this number, the information about communication may be lost for the researcher, and audio or video recording may be superior.

### 3.3. Quantitative Data Processing

The following data may be extracted from the temporal interaction diagrams. A qualitative analysis yield results similar to the coding of videos. The quantitative analysis gives frequencies of different types of interactions.
Total meeting time and time spent on every slide (observed communication part).Distribution of the main team roles among participants

Communication events at the meeting were divided into two main groups: sending information and receiving information. The first group includes: *addressing*—appealing to a particular person; *transmitting*—appealing to the whole group or commenting without particular addresser; *information providing*—answering questions; and *artefact providing*—showing artefacts for an explanation. The second group is *receiving information* and may include answers or any other addressed information from the participants of the meeting.

A point system was devised to show the contribution of every participant to the possible type of main team interaction:*Addresser*: initiates two or more interpersonal interactions, excluding artefacts*Transmitter*: comments three or more times (circle interactions on diagram)*Information Provider*: provides two or more answers*Artefact Provider*: shows any new artefact (e.g., models on paper, electronic models, physical objects, and presentations)
✓SENDER: This is the sum of the points for outcoming information (addresser, transmitter and provider roles)✓RECEIVER: Receives two or more addressing interactions including answers✓OUTSIDER: Has fewer than two interactions (any)

Therefore, the scoring determines the number of interactions per slide, against each of the above interaction types. These are summed into SENDER and RECEIVER categories, or OUTSIDER if the member had few interactions of any type. Note that these categories apply only to the slide in question, and in the next time period a group member can change their interaction.

These slide points were summed up to receive a final result indicative of the distribution of roles for each participant during the meeting. The proportion of the main three parts (sender, receiver and outsider) show the prevailing type of communication interaction in the team.

Example:

The example of the role distribution among participants is shown in Table 1. This group included many participants (the number was different at every meeting). At the first meeting, there were no supervisors among students. The following data are for the whole meeting, i.e., multiple slides. The letters A–F indicate members of the group. The numbers indicate the quantity of a particular type of interaction.

Table 1 shows that Participant A received seven points for communicating in the role of *Sender* (four points for transmitting information, one point for addressing to other members of the team and two points for providing answers). In general, the role of *Sender* was taken by Participants A, C, D and F (assuming “take role” means have twice as many points as for any other column), whereas Participants B and E had a wider distribution of roles.

3. The density of communication events for each participant

Communication interaction can be understood as a change of turn-taking among participants or change of conversation addresser. The number of such interactions during the meeting was calculated and used to find the density (interactions per minute) of communication events for each participant.

Example:

This example refers to the first meeting of the Group 1. Table 2 shows the calculation of the density of communication events for each participant and the average density per meeting.

From the results in Table 2, the most active team members at the meeting were A and F. Total time spent for this meeting was 46 min. Total group activity was 2.48 interactions per minute. The coefficient of variation between slides exceeds one (CV > 1) only with Participant B (high variability). That suggests that this member of the team changed their behaviour during the project discussion more often than other participants.

4. Total group activity 

Total group activity can be determined from the total quantity of interactions divided by total meeting time. These indicators can also be compared between meetings.

Example:

The example of total group activity is shown in Table 3. The density of communication events for all seven meetings of Team 1 were compared.

In our case, we cannot compare communication activity between meetings, because, according to the ethics agreement, a participant’s identity could not be followed and recorded through the meetings. Therefore, identification codes of people were different every time. However, in other situations, it could be interesting to make such a comparison and to follow communication activity of every member across the project completion time. Figure 4 illustrates the density of communication events at different meetings of Team 1.

In Figure 4, the project team has maximum communication activity in the sixth meeting. To understand why this happened, it is necessary to conduct a qualitative analysis of the same meeting.

As shown above, the interaction diagram method has the ability to be analysed quantitatively. The next section demonstrates its qualitative capabilities.

### 3.4. Qualitative Analysis

#### 3.4.1. Analysis of Repeating Patterns of Communication Behaviour

The method offers the possibility to observe a group in development through different meetings, by extracting data about typical group activity (e.g., slow communication processes, intensive communication, or many discussions in parallel). In addition, it is possible to analyse the individual behaviour of the team members—repeating patterns of communication events or long monologue interactions.

The research journal can be written on the basis of information extracted from the ID diagram. Some words or phrases representing behaviour of team members can be underlined and used later to identify the team roles.

#### 3.4.2. Example of Data Extraction and Written Notes

Six members of Team 2 came to the first meeting: one supervisor, one assistant (postgraduate student) and four students (undergraduates). The data below were extracted from the diagrams (11 slides in total) and later described in the research journal. In addition, the keywords were identified and underlined there.

In this meeting, we observed two particularly active communication couples. The first was supervisor F and student C. The student asked questions to a supervisor, gave answers and provided specific information when others could not. Supervisor F communicated a lot with the participant C and with other students, defined tasks in the beginning and at the end. 

Another communication couple was assistant E and student C. Student C listened to the assistant E carefully, consulted with him, asked many questions. Assistant E often provided information when others asked him (including the supervisor). Apparently, that was not only because of his higher official position than undergraduates but also because of high status in the team (specific team role). Student B was very passive, did not ask a question or participate in the discussion. 

According to the observational notes, participant D (undergraduate student) typically agreed with others (verbally and non-verbally) and sometimes participated in the discussion by asking a few questions (mostly to student E), but not actively. As opposed to B and D, participant A was always active during the meeting, answered to the questions of other group members, however he apparently preferred referring comments and explanations to the whole team rather than appealing to a particular team member. In addition, participant A expressed group feeling several times, summarised ideas.

The dynamics of communication changed significantly over the time of this meeting: the F-C communication couple was more active at the beginning of the project discussion. In the end, it became *quite passive*, only asking questions to each other. The E-C communication couple, on the contrary, increased its communication activity towards the end of the discussion time.

In addition, in the middle of the meeting time, assistant E showed a diagram on the computer to the supervisor F, and this changed group communication dynamics. Everybody then started listening and watching closely to this participant and his interlocutor (supervisor, however, added something, asked questions and collected information from students). Two students (the one from the first couple C and passive B) did not show any signs of interest in this model. It is possible that the long physical distance from the provider of the artefact (model) may have contributed.

During the last few minutes of project discussion the density of communication events was equally distributed between participants, with the exception of the consistently passive student B.

In addition to the keywords that were identified and underlined in the journal, there are also groups of keywords (or communication patterns) that came directly from the diagrams. They are written on the ID near the person that talks or shows non-verbal signs and may change from slide to slide, for example see Figure 3 (Person B, text “suggest”). Non-verbal behaviour can be shown in diagrams using dotted-lines (e.g., attention to the written model on the whiteboard, gestures, and smiles). However, these interactions are challenging to record as generally they happen in parallel with verbal ones.

#### 3.4.3. Identification of Roles

We devised a categorisation for team roles. To do this, we noted the team roles identified in [36,37]. We then added the roles that we observed over multiple interactions. These were based on the above keywords, which we grouped into common themes. In this way, we created a set of team roles, with associated communication patterns (see Table 4). We do not claim that this categorisation is validated.

Key differences between this categorisation of team roles and the literature [37] are the use of more than six team roles to explain participants’ behaviour. Some team roles were elaborated and changed (e.g., *Innovator* was turned into *Initiator* because our observation is that initiating of conversation does not always lead to generation of new ideas), other roles were considered but abandoned. For example, *Challenger* [37] was described as someone who pushes the team to solve the problems and find different solutions. In our case, the nature of communication suggested *Initiator* or *Facilitator* were a better description of the behaviour we observed. 

Most of the roles below came from [36], but were adapted to the engineering project team environment by the authors. We also considered the practicality of identifying these roles using our diagrammatic method, i.e. the measurability. Thus, *Explorer* (ask questions, ideas and opinions) is the simplified version of *Information seeker* (ask for clarification, suggestions and facts pertinent to the problem) [36]. We found that the role of *Explorer* can be identified from the IDs even after several months after observations, whereas the content of speech cannot be recorded so easily.

The *Gatekeeper* role came directly from [36] without changes. Other team roles from [36] were skipped (Individual roles) or combined (*Compromiser* and *Harmoniser* were joined into *Arbitrator*). We also suggest new roles that characterise the nature of discussions: *Connector*, a person who connects the project team with other groups outside the organisation; *Facilitator*, elected Team Leader at particular meeting (not necessarily supervisor or manager); *Representative*, a person who talks on behalf of others in the team and thus represents group ideas or feelings (for example, one student reported to the supervisor about the team achievements or answered their question); and *Outsider*, a person in group who does not participate in the discussion, which is similar to *Playboy* in [36] but without the negative connotation.

Many other team roles could be envisaged, but this may make the process of identifying such roles overly complicated. The ten team roles suggested in Table 4 are what can be easily observed and identified while letting the observer simultaneously write other notes and build diagrams. In addition, these team roles can be correlated with the “main team interactions” described above, hence the quantitative part of the observation can be correlated with the qualitative.

**Example of team roles:** In our example (first meeting of Team 2), this qualitative analysis helped to identify the following team roles at the first group meeting:Participant A—*Information provider* (providing detailed and excessive information) and *Representative* (verbalising group’s feelings, providing an answer to the question that referred to all group)Participant B—*Outsider* (passive communication behaviour, almost did not participate in project discussion)Participant C—*Information provider* (providing detailed and excessive information), and *Explorer* (asking many questions)Participant D—*Passive collector* (non-verbal signs of agreement or just short yes/no answer, low verbal participation in team discussion, attentive listening)Participants E—*Information provider* (providing detailed and excessive information)Participant F—*Facilitator* (defining the task or group problem)

As this shows, participants may have several team roles. In addition, some roles may intersect with each other, and it can be difficult to define them.

### 3.5. Combination of Observational Data

Quantitative and qualitative methods can be combined for broad analysis of team communication (see Table 5 and Table 6). This shows that the team roles and interactions identified by the methods are similar.

## 4. Discussion

### 4.1. Practicality of Observation and Preparation

We noted that behaviour of students who had a meeting in the supervisor’s office was quite different from those meeting in neutral territory. From this, we infer that the place of observation may be important and influence the behaviours. We found it useful if there was a quiet place or corner in the room where the observer could sit and make notes without being intrusive to the meeting. This place, however, should give the observer opportunity to see participants’ interaction. i.e. not only hear the speech but also see non-verbal behaviours.

The other difficulty of observation setup is a preparation of materials. In our experience, the choice of paper may be important. Thin paper produces noise that disturbs the meeting. A small format notebook makes the observer turn pages frequently, during which some details of communication can be missed. A suitable medium was found to be a notebook of A5 size with pale white pages. The ink also should be chosen carefully (bleeding or pale ink makes diagram notes hard to read). The flow chart in Figure 5 shows the preparation for experiment.

This ID method does not need special training and can be used by different observers (researchers or managers in the organisation). However, the group size should not exceed about eight people, and all participants should sit or move only a little during the observation. Otherwise, moving, intensive talking, and parallel discussions that sometimes appear in the big group may create difficulties in note-taking (errors and missed turn-taking). In this case, video-recording may be better.

In addition, knowing participants in person as quickly as possible increases chance for better data collection. When observers start their work at each meeting, they need first to recognise the participants and locate points (nodes) with letters (or coded names) on the paper that correspond to people in the room. Each meeting position of participants can be different, so it is important to do the setup quickly and not lose initial information. 

After that, each new slide usually has the same position of nodes on the diagrams (provided that people do not move), so the observers’ work becomes easier and consists of quick note-taking and line-drawings. When observers feel the data on the slide are enough, they draw nodes on another page in the notebook, thus preparing for the next slide. It takes seconds and does not have big impact on the process of observation. Here, it is important not to forget to fix the beginning time of each slide if observation is conducted for research purposes, otherwise for the purpose of commercial observation that may not be needed.

### 4.2. Advantages of the Method

We have devised a method that may be used to capture major interactions within teams. This method allows low-invasive observation and recording of multiple interactions between participants: time sequence, direction and type of interactions, using of artefacts, participants’ characteristics and group roles. This was achieved without requiring a recording device, other than a paper notebook. In turn, this makes ethics approval much easier. The method can also be less invasive and inhibiting for participants. It can be used in a time-pressured situation, within limits. A further advantage is the method obviates the need for post-event data transcribing, which is otherwise an onerous task for the researcher. 

### 4.3. Point of Difference

Points of difference compared to standard sociograms:Allows recording time sequenceShows direction of member-to-member interactionsAllows recording specific situational behaviour of different team membersRecords use of artefacts by participantsAllows recording non-verbal behaviour, but only for a limited period of timeShows long monologue interactionShows repeating patterns of the interactions between same group members

Points of difference compared to audio and video recording:Several interactions, e.g., non-verbal agreements, nodding or gestures would not have been detected with audio, but were captured with the ID method. With audio recording, there can also be identification problems with multiple people speaking at once, which is less of an issue with the ID method.Video could pick up all these and has the additional advantage of being able to be re-played. However, video recording changes the behaviour of participants, and requires more stringent ethics approvals.

### 4.4. Domain Specific and Generic Elements

Domain specific elements in this study were the engineering nature of the work. The purpose of the meetings in this case was engineering problem-solving, hence the nature of the interaction was directed to task progression. In other contexts of human meetings, the nature of the interaction can be expected to be different. The symbols we developed were for the engineering context, and both these and the structure of the interactions may need to be revised in other situations. Nonetheless, we suggest the following elements are generic, and might be expected to appear in multiple areas: *the starting time; numbers*, the sequence of communication interactions; *letters*, participants of the meeting; *circles*, broadcast speech; *arrows*, the direction of communication; *question signs*, questions; *small arrows* near question marks, answers and repeating questions; *notes* written near participant’s letter; *solid line* (verbal communication interaction); and *dotted line* (non-verbal). 

### 4.5. Limitations of the Interaction Diagram Method

The method is limited to observation of small- or medium-sized groups (maximum about eight people) because of the manual nature of the recording. It is difficult to record the simultaneous non-verbal behaviour of multiple members, or if members constantly move about in the meeting. In addition, this method involves researcher’s judgements about what level of detail to choose (e.g., any interactions or only verbal ones), what to consider an artefact, personal interpretation of situation (e.g., differentiating transmitting from addressing), and data presentation (how to represent new events). These limitations are similar to transcription [1]. The “observer effect” [2] still exists because of the presence of the researcher. Another limitation is that the method does not provide a written verbatim transcription. 

The method has not been directly compared to video recording. It would be interesting to determine whether some interactions might be missed, that might be detectable from video recording. It is to be expected that the observer might miss interactions during busy discussion periods, or in meetings with many active participants. Our initial observation from experience is that not keeping up with the interactions adversely affects the quantitative analysis but is not so damaging to the qualitative analysis. Other action communication situations as crew environments, or construction and operational activities, may require full video recording.

### 4.6. Application

This method was designed primarily for researchers who need to observe group interactions between team members in an engineering organisation/university without audio or video recording. However, it could also be used by managers of organisations, for example as a supplement to minutes. Other possible applications include: the qualitative part of the ID method might be used for team formation or team recruiting, while the quantitative part might be used for appraisal and performance review. However, we note that the quantitative analysis is time-consuming and may be better for research purposes rather than commercial application.

### 4.7. Implications for Further Research

This observation was exploratory by nature because of its short-term longevity and limitations (not able to follow participants between meetings and no questions asked during meeting). The purpose was to develop a novel method of observation. A possible future research project might be to validate the method by comparing video-recording of the meeting and note-taking. This has the potential benefit of increasing the objectivity of the empirical study.

Another line of enquiry could be to further develop the method by including: non-verbal interactions (develop a way to show non-verbal interactions in parallel with verbal ones); artefact abbreviations (create a list of possible artefacts and their abbreviations); and many people in the meeting room (improve method so it can show interactions of many participants).

Further research ideas could be to explore specific aspects of role assumption. The exploratory study has shown that formal group role (supervisor) does not necessarily lead to the corresponding leading team role (*Facilitator*, *Arbitrator*, etc.). Sometimes these roles are taken by a student in a group. Why is this? A similar question is whether the choice of artefacts and physical location in the room are predefined by the team roles, formal status or by personality traits.

Our hypothesis was that participants would be less inhibited by this method than video/audio recording. While this did appear to be so, we could not validate this during the present study. Hence, it could be interesting to use multiple researchers for objectivity, and measure the “observer effect” [2].

## 5. Conclusions

The objective was to develop a method for team observations. The resulting interaction diagram method is capable of representing multiple interactions between participants in a time-pressured situation in time sequence, and it provides a means to quantify the number and type of personal communication interactions. 

The work makes the following original contributions. First, it provides a graphical representation of the record of the interaction flow during meetings. It does this without needing video recording. It is also an efficient method, as it does not require subsequent transcription or coding. Second, it provides a procedure to quickly analyse communication situations, identify group roles, and compare group activity at different meetings.

## Figures and Tables

**Figure 1 behavsci-09-00005-f001:**
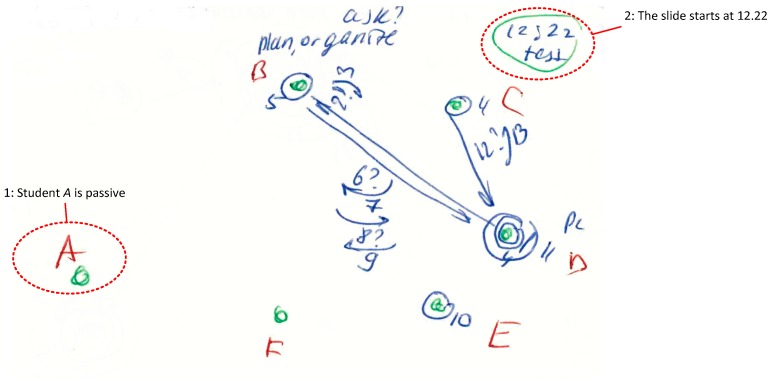
Interaction diagram: Communication Situation 1. See text for explanation.

**Figure 2 behavsci-09-00005-f002:**
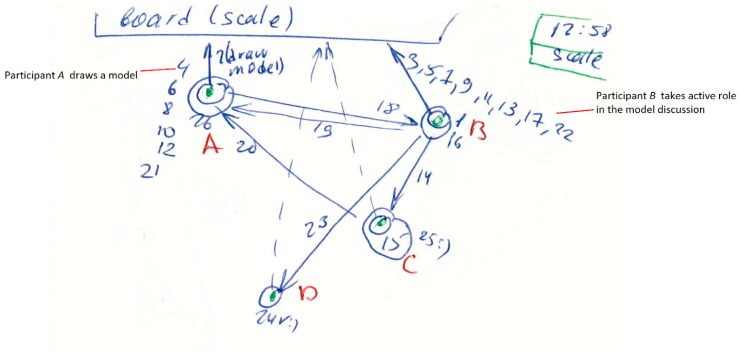
Interaction diagram: Communication Situation 2.

**Figure 3 behavsci-09-00005-f003:**
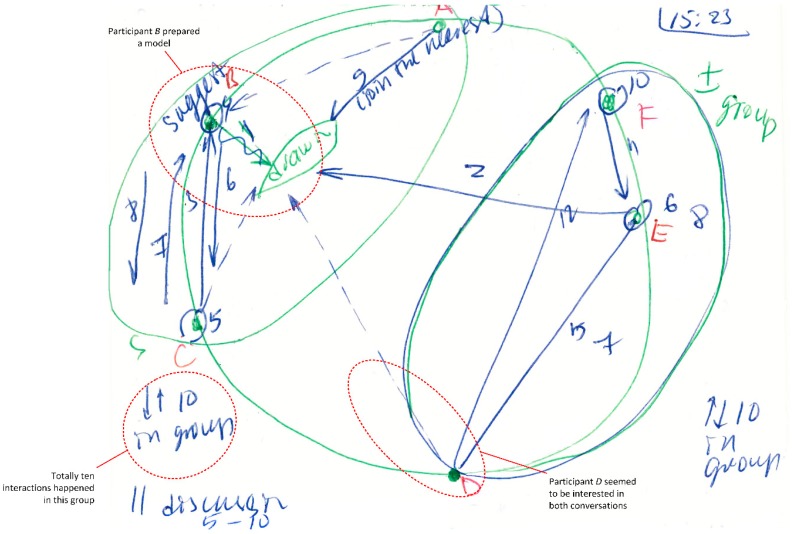
Interaction diagram: Communication Situation 3.

**Figure 4 behavsci-09-00005-f004:**
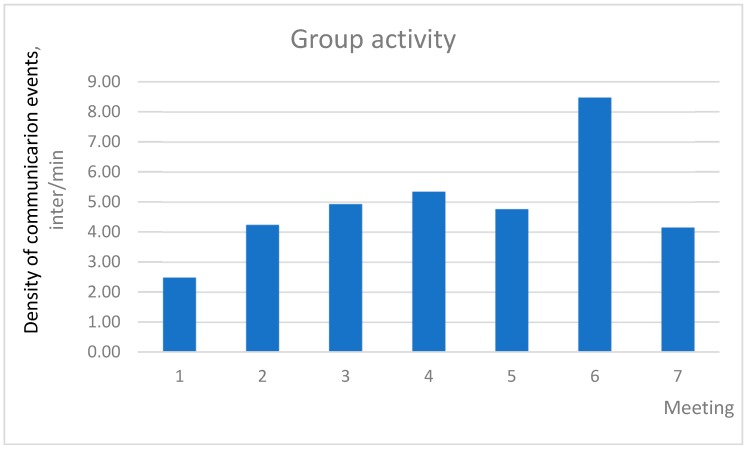
Group activity of Team 1 through Project Meetings 1–7.

**Figure 5 behavsci-09-00005-f005:**
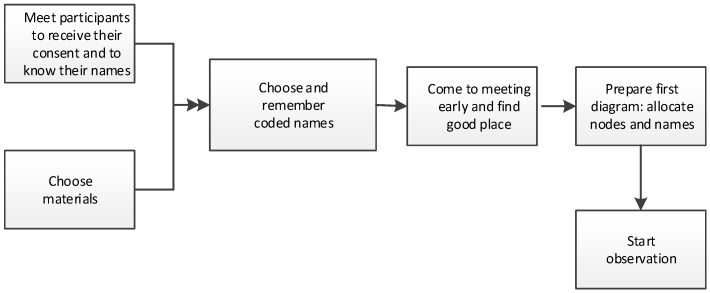
Setup of the observation.

**Table 1 behavsci-09-00005-t001:** Distribution of the main team roles among participants of Group 1 at first meeting.

Addresser	Transmitter	Information Provider	Artefact Provider	Sender (sum)	Receiver	Outsider
A	4A	2A	-	7A	2A	-
-	2B	-	-	2B	-	3B
3C	2C	-	-	5C	C	2C
-	4D	-	-	4D	-	-
-	4E	-	-	4E	3E	1E
4F	2F	F	F	8F	-	1F

**Table 2 behavsci-09-00005-t002:** The density of communication events in Group 1 at the first meeting.

Slide	Time, Min	Quantity of Interactions, Per Person							Density of Communication Events, Interactions Per Min					
		A	B	C	D	E	F	Total	A	B	C	D	E	F
**1**	17	4	2	5	3	3	1	**18**	0.24	0.12	0.29	0.18	0.18	0.06
**2**	7	3	4	4	3	3	6	**23**	0.43	0.57	0.57	0.43	0.43	0.86
**3**	5	7	2	1	1	2	4	**17**	1.40	0.40	0.20	0.20	0.40	0.80
**4**	6	5	1	4	3	5	5	**23**	0.83	0.17	0.67	0.50	0.83	0.83
**5**	5	5	0	1	4	0	4	**14**	1.00	0.00	0.20	0.80	0.00	0.80
**6**	6	3	0	3	6	5	2	**19**	0.50	0.00	0.50	1.00	0.83	0.33
**Total**	46	27	9	18	20	18	22	**114**						
**Density for every participant * (arithmetic mean)**									**0.73**	**0.21**	**0.41**	**0.52**	**0.45**	**0.61**
**Density for every participant * (average per meeting)—total quantity of interactions divided on total time**									**0.59**	**0.20**	**0.39**	**0.43**	**0.39**	**0.48**
Standard deviation between slides (for all group)									0.43	0.23	0.20	0.33	0.34	0.34
Standard deviation between participants (from arithmetic mean)									SD = 0.18					
Standard deviation between participants (from average)									SD = 0.13					
Coefficient of variation CV (slides)									0.59	**1.10**	0.5	0.63	0.76	0.55
Coefficient of variation CV (participants) from the arithmetic mean									0.37					
Coefficient of variation CV (participants) from average									0.31					
**Total group activity—2.48 inter/min**														

* This can only be determined within one meeting unless the identity of the participants is preserved from meeting to meeting.

**Table 3 behavsci-09-00005-t003:** The density of communication events of Team 1.

Meeting	Total Time, Min	Group Communication Activity, Interactions Per Minute [Mean]	Standard Deviation SD and Coefficient of Variation CV between Meetings
1	46	2.48	
2	55	4.24	
3	42	4.93	SD = 1.82
4	50	5.34	CV = 0.37
5	49	4.76	
6	58	8.48	
7	52	4.15	

**Table 4 behavsci-09-00005-t004:** Suggested correlations between communication patterns and team roles.

N.	Team Roles	Typical Communication Pattern
1	*Initiator* (initiate process)	Active participation, propose new ideas and tasks, new directions of work.
2	*Passive collector* (collect information)	Passive data collecting, non-verbal signs of agreement or just short yes/no answer, low verbal participation in team discussion, attentive listening, and keeping ideas inside (non-vocalisation).
3	*Explorer* (ask questions)	High verbal participation, active data collecting: ask general questions, ask for different facts, ideas or opinions, and explore facts. Ask to clarify or specify ideas, define the term, and give an example.
4	*Information provider* (provide information)	Provide detailed and excessive information: take an active part in the conversation, but mostly talk rather than listen.
5	*Facilitator* (summarise, control discussion)	Define the task or group problem, suggest a method or process for accomplishing the task, provide a structure for the meeting, control the discussion processes. Bring together related ideas, restate suggestions after the group has discussed them, offer a decision or conclusion for the group to accept or reject. Get the group back to the track.
6	*Arbitrator* (solve disagreement)	Encourage the group to find agreement whenever miscommunication arises or group cannot come to a common position.
7	*Representative* (express, answer)	Verbalise group’s feelings, hidden problems, questions or ideas that others were afraid to express, provide an answer to questions that were referred to the whole group.
8	*Gatekeeper* (fill gaps, sensitive to others)	Help to keep communication channels open: fill gaps in conversation, ask a person for his/her opinion, be sensitive to the non-verbal signals indicating that people want to participate.
9	*Connector* (connect people)	Connect the team with people outside the group.
10	*Outsider* (stay outside)	Do not participate in project discussion.

**Table 5 behavsci-09-00005-t005:** Combination of observational data for the team roles and interactions.

Participant	Qualitative Team Role	Main Team Interactions (See Table 1)
A	*Information provider*	*Sender*
B	*Outsider*	*Outsider/Transmitter*
C	*Information provider*	*Sender*
D	*Passive collector*	*Transmitter*
E	*Information provider*	Not defined (equal distribution)
F	*Facilitator*	*Sender* (very active)

**Table 6 behavsci-09-00005-t006:** Combination of observational data for the group activity.

Meeting No.	Qualitative Analysis of the Total Group Activity (Notes from Observation)	Quantitative Analysis of the Total Group Activity, (Interactions per Minute)
1	The team members defined their tasked and goals. The middle communication activity	2.48 (middle)
2	The dynamics of communication was at the low level initially and then increased towards the end of the meeting.	4.24 (high)
3	The communication activity was rather high. There were nine people in the room. They all talk at the same time, and there were many discussions happened in parallel.	4.93 (high)
4	The communication activity was very high. There were nine members of the team again. However, discussions in parallel were not observed. The appearance of boundary objects (computer model) intensified the communication strongly in the middle of the meeting.	5.34 (above high)
5	The communication activity was high even if there were only four participants at the meeting. First, one of the students prepared the physical model, which was very intensively discussed. Then, another student explained his ideas on paper charts. Later, the third student showed video-presentation. That attracted big attention and caused an intensive wave of discussion again.	4.76 (high)
6	The communication activity was extremely high. Discussion started intensively from the very beginning and continues until the end of the meeting. Participants talked at the same time, and there were many discussions happened simultaneously. The students and supervisors discussed the submission of the project proposal during the next week. Team members also used artefacts for explanations. In general, it was hard to record the communication in the team because of the high speed of turn-taking, and many discussions happened in parallel.	8.48 (very high)
7	The discussion was not very intensive at the beginning and the end but revived in the middle when the supervisor came into the room. There were some parallel discussions only over the last 5 min of the meeting. The project proposal had already been submitted.	4.15 (high)

The results show that the quantitative and qualitative methods complement each other. In our case, we defined communication activity below 2 as “low activity”, between 2 and 4 as “middle”, 4–5 as “high”, 5–6 as “above high” and more than 6 as “very high”. These thresholds were determined based on the range observed, which ran from 2 to 9. It is acknowledged that these thresholds are subjective.

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
