# Peer review of "Interaction Diagrams: Development of a Method for Observing Group Interactions"

_behavsci, 2018, doi:10.3390/bs9010005_

Round 1

Reviewer 1 Report

The paper presents a method for "recording" on paper group interactions.

The paper reads nice and my only concern is about its apparent novelty. Admittedly, I am not an exert on the field, but I cannot imagine that tools similar to the proposed interaction diagrams have not been used before (given that sociogram are very old and the extensions proposed seem quite natural). On the other hand I have no evidence for the opposite (given that I am from a different field). So I propose that the authors epand their coverage of tools for recording to include other manyual/paper-based recording tools (a sthey cover electronic/software tools for recording).

I would also welcome some more discussion on the ease (or difficulty) in producing these diagrams. The authors address this issue in the limitations but it would be nice to understand beeter e.g. how mauch training is reuired to get a certain fluency in producing ID (e.g. for big groups of 6-8 persons).

Finally, I would like to see a dixcession on how the results are affected by errors and/or missed (i.e. unaccounted) interactions. I guess it is natural for the observer to miss some ineraction (especially in a fast paced discussion). I also guess that such mishaps will not change much the results of the the overall "recording". However I would like to see if this can be somehow modelled and quantified to etsimete the "resiliency" of the method to erros.

Some minor comments:

line 59: "(n=519, Japanese companies) " probably refers to some info about [4[ but is out of context here.

line 210: I cannot locate the brackets that indicate the repeating patterns in fig.3. I can see up/down arrow with numbers next to them, but no brackets.

line 360: Please indicate what is high activity (e.g. intensity>2 or something like that), as all metings in fig. 4 are said to have high activity with no indication of what is high and what is not.

line 351 and line 419: the caption of tables 3 and 4 are in different pages of the tables themselves.

Author Response

Reviewer 1

The paper reads nice and my only concern is about its apparent   novelty.

Admittedly, I am not an expert on   the field, but I cannot imagine that tools similar to the proposed   interaction diagrams have not been used   before (given that sociogram are very old and   the extensions proposed seem quite natural).   On the other hand, I have no evidence for the opposite (given that I am from   a different field). So, I propose that the authors expand their coverage of   tools for recording to include other manual/paper-based   recording tools (as they cover electronic/software tools for recording).

Many works have been devoted   to the communication mapping in 40-60s. The concept of communication   patterns, which is similar to sociograms, uses lines and symbols. H. J.   Leavitt [1] in 1950 identified and diagrammed typical patterns of the   communication (wheel, Y, chain, circle and network). These patterns and the   position of the paricipants in it were correlated then with behavioral   differences of people [1].

In most of these works the communication   pattern was defined as connectivity between nodes, e.g.  [1,2], and so the diagrams were evaluated quantitatively (number   of connections, symmetry etc.). The communication was presented as an   exchange of messages between participants through channels, i.e. a   sender-receiver model.

In contrast, the method shown in the current   paper differs by using diagrams as a platform for qualitative data (although   quantitative data can be extracted too). We are interested not so much in the   relationship-structure of the group at meeting, nor the position of the team   members in the communication network, but rather how people react to others’ behaviour,   what words and non-verbal signs are used, what proceeds the communication   events at the meetings, and how different situations (e.g. parallel   discussion, use of artifacts, appearance of another new person) change the   communication environment in the group. In this sense we less interested in   the communication pattern, and more in the communication behaviors. The ID   method provides a mechanism to represent these behaviours by using diagrams   with time-line.

Text to this effect has been added   to the manuscript, see section 1.

I would also welcome some more   discussion on the ease (or difficulty) in producing these diagrams. The   authors address this issue in the limitations, but it would be nice to   understand better, e.g. how much   training is required to get a certain fluency in producing ID (e.g. for big   groups of 6-8 persons).

Thank you for the good suggestion.   Details of preparation and difficulties of using this method were added to   the manuscript, see new section 4.1

Finally, I would like to see a   discussion on how the results are affected by errors and/or missed (i.e. unaccounted) interactions. I guess it is   natural for the observer to miss some interaction (especially in a fast-paced   discussion). I also guess that such mishaps will not change much the results   of the overall "recording". However, I would like to see if this   can be somehow modelled and quantified to estimate the "resiliency"   of the method to errors.

The method has not been directly compared to video recording. It   would be interesting to determine whether some interactions might be missed, that might be detectable from the video recording. It is to be expected that the observer might miss   interactions during busy discussion periods, or in meetings with many active   participants. Our initial observation from experience is that not keeping up   with the interactions adversely affects   the quantitative analysis but is not so damaging to the qualitative analysis.  

Text to this effect has been added to the manuscript in section 4.5.

Some minor comments:

line   59: "(n=519,   Japanese companies)" probably refers to some info about [4] but is out   of context here.

The sentence has been edited to be   clearer.

line 210: I cannot locate the brackets that   indicate the repeating patterns in fig.3. I can see up/down arrow with   numbers next to them, but no brackets.

You are correct. What we meant was   ‘as a separate group’. Manuscript corrected.

line   360: Please   indicate what is high activity (e.g.   intensity > 2 or something like that), as all meetings in fig. 4 are said   to have high activity with no indication of what is high and what is not.

Noted.  The meetings in fig.4 are of different   communication activities and cannot be   identified as ‘high intensity’. The scale for communication activity was presented after the Table 6.

In   our case, communication activity below 2 we defined as ‘low activity’,   between 2 and 4 – ‘middle’, 4-5 – ‘high’, 5-6 –‘above high’ and more than 6 –   ‘very high’. These thresholds were determined based on the range observed, which   ran from 2 to 9. It is acknowledged that these thresholds are subjective.

See   new text in section 3.5.

line   351 and line 419: the caption of tables 3 and   4 are in different pages of the tables themselves.

Edits have been made to prevent   this. This will hopefully also be addressed in the copy setting.

1.            Leavitt,   H.J. Some effects of certain communication patterns on group performance. The   Journal of Abnormal and Social Psychology 1951, 46, 38.

2.            Shaw,   M.E. Communication networks. In Advances in experimental social psychology,   Elsevier: 1964; Vol. 1, pp 111-147.

3.            Benne,   K.D.; Sheats, P. Functional roles of group members. Journal of social issues   1948, 4, 41-49.

Reviewer 2 Report

I would like to thank the authors for discovering and sharing an innovative approach to systematic note-taking. I believe a few concerns should be addressed before possible publication.

First, the authors seem to shift between describing this approach as a way to take notes for organizations (perhaps as a supplement to minutes) and for research purposes. I think that both uses might be appropriate. The authors should consider having separate sections for each.

The small group communication literature has a long history of mapping in a manner similar to yours. Leavitt (1951) identified and diagrammed wheel, chain, y, circle, and all-channel orientations. Shaw (1964) did a summary of 18 studies looking at network type and problem-solving effectiveness. I would encourage you to investigate this body of literature.

Of course, as the group size increases arithmetically, the possible interactions increase geometrically. I am concerned that the efficacy of the maps might be limited by the group size. 

The number of interactions over time should be called density, not intensity. Intensity usually refers to some special emphasis placed on a comment or question.

I am not sure that the quantitative analysis tells us anything. Why would we want to know the density of comments broken down by group member? I think you should present some contexts where this information would be helpful.

Again, I do think that the approach is interesting. I am not sure that we are learning anything entirely new. Perhaps with more discussion about application and implications, I would be more willing to support publication.

Author Response

Reviewer 2

I would like to thank the authors   for discovering and sharing an innovative approach to systematic note-taking.   I believe a few concerns should be addressed   before possible publication.

Thank you for this positive   comment.

First, the authors seem to shift   between describing this approach as a way to take notes for organizations   (perhaps as a supplement to minutes) and for research purposes. I think that   both uses might be appropriate. The authors should consider having separate   sections for each.

Originally, the method was   developed to help researchers in data collection without recording tools. However,   taking notes can be used also by organisations for practical purpose. A new   section has been added as section 4.6 to address this.

The small group communication   literature has a long history of mapping in a manner similar to yours.   Leavitt (1951) identified and diagrammed wheel,

chain, y, circle, and all-channel   orientations. Shaw (1964) did a summary of 18 studies looking at network type   and problem-solving effectiveness. I would encourage you to investigate this   body of literature. 

Many works have been devoted   to the communication mapping in 40-60s. The concept of communication   patterns, which is similar to sociograms, uses lines and symbols. H. J.   Leavitt [1] in 1950 identified and diagrammed typical patterns of the   communication (wheel, Y, chain, circle and network). These patterns and the   position of the paricipants in it were correlated then with behavioral   differences of people [1].

In most of these works the communication   pattern was defined as connectivity between nodes, e.g.  [1,2], and so the diagrams were evaluated quantitatively (number   of connections, symmetry etc.). The communication was presented as an   exchange of messages between participants through channels, i.e. a   sender-receiver model.

In contrast, the method shown in the current   paper differs by using diagrams as a platform for qualitative data (although   quantitative data can be extracted too). We are interested not so much in the   relationship-structure of the group at meeting, nor the position of the team   members in the communication network, but rather how people react to others’ behaviour,   what words and non-verbal signs are used, what proceeds the communication   events at the meetings, and how different situations (e.g. parallel   discussion, use of artifacts, appearance of another new person) change the   communication environment in the group. In this sense we less interested in   the communication pattern, and more in the communication behaviors. The ID   method provides a mechanism to represent these behaviours by using diagrams   with time-line.

Text to this effect has been added   to the manuscript, see section 1.

Of course, as the group size   increases arithmetically, the possible interactions increase geometrically. I   am concerned that the efficacy of the maps might be limited by the group   size.

You are right. There is a   limitation in using this method: maximum 6-8 people in the meeting. However,   the complexity of observation depends not only on the quantity of   participants but also on their communication and physical activity. Eight   silent and sitting at the desk people in the room still can be easily   recorded by note-taking, but six moving and talking people can be a hard   task.

This is addressed in new section   4.1.

The number of interactions over   time should be called density, not intensity. Intensity usually refers to   some special emphasis placed on a comment or question.

Thank you for the helpful comment.   Density is more appropriate in our case as it refers to the frequency of   events. The text has been changed.

I am not sure that the   quantitative analysis tells us anything. Why would we want to know the   density of comments broken down by group member? I think you should present   some contexts where this information would be helpful.

Agreed. The Quantitative part of this study has limited   application. We suggest that it is used primarily for research purposes as it   is time-consuming. See new text in section 4.6.

Again, I do think that the   approach is interesting. I am not sure that we are learning anything entirely   new. Perhaps with more discussion about application and implications, I would   be more willing to support publication.

The application part of the paper has been  extended with new section 4.1 to address   this.

1.            Leavitt,   H.J. Some effects of certain communication patterns on group performance. The   Journal of Abnormal and Social Psychology 1951, 46, 38.

2.            Shaw,   M.E. Communication networks. In Advances in experimental social psychology,   Elsevier: 1964; Vol. 1, pp 111-147.

3.            Benne,   K.D.; Sheats, P. Functional roles of group members. Journal of social issues   1948, 4, 41-49.

Reviewer 3 Report

Please find comments in attached file.

Author Response

Reviewer 3

Thank   you for thought-provoking questions.

A   flow chart was added to the manuscript,   see Figure 5.

The   work is primarily conceptual in nature, demonstrated with real data. As such   the method has limitations, which we have expanded on in the extensively   modified discussion (Section 4).

Most of the roles below came from [3], but were adapted to the   engineering project team environment by the authors. We also considered the   practicality of identifying these roles using our diagrammatic method, i.e.   the measurability.

We   do not claim that this categorisation is validated.

Domain   specific   elements in this study were the engineering nature of the work. The purpose   of the meetings, in this case, was engineering problem-solving, hence the   nature of the interaction was directed to task progression. In other contexts   of human meetings, the nature of the interaction can be expected to be   different. The symbols we developed were for the engineering context, and   both these and the structure of the interactions may need to be revised in other situations. Nonetheless, we suggest that certain elements   may be generic.

See   new section 4.4 that addresses this question.

Good point and a number of changes have been made to improve consistency